# OpenReview forum: "Building a Foundational Guardrail for General Agentic Systems via Synthetic Data"
_ICLR.cc/2026/Conference — ICLR 2026 Poster_

### Official Review · Reviewer_xJJZ · 2025-10-27

**Soundness:** 3
**Presentation:** 4
**Contribution:** 3
**Rating:** 8
**Confidence:** 2

**Summary:**

This paper aims to improve the safety of LLM agents by intervening during the "planning stage", before the agent executes the plans. The manuscript highlights three gaps in the current literature: the data gap, a model gap, and an evaluation gap. The paper proposes a contribution for each.

To close the data gap, the authors propose AuraGen, a synthetic data generation framework that introduces unsafe perturbations to safe plans, thereby creating a training corpus for planning-stage LLM agent safety approaches. AuraGen features four injection modes: Single-step perturbation, Multi-step corruption, New-branch diversion, and Bridged-branch diversion, ensuring a diverse set of unsafe plans can be generated.

To close the model gap, the authors propose Safiron, which features a unified adapter module that normalizes inputs and a guardian model that outputs a classification (plan: safe vs. unsafe), a risk categorization, and a textual explanation. Safiron is trained by supervised fine-tuning on AuraGen data, followed by RL (GRPO) to further optimize output behavior.

Lastly, to close the evaluation gap, the authors propose Pre-Exec Bench, a benchmark for evaluating the safety of agentic plans before execution.

The results show that AuraGen generates a balanced dataset featuring different kinds of risks. The proposed model, Safiron, is shown to outperform almost all existing models on classification accuracy, harmful detection precision, risk categorization, and explanation correlation.

---
Note: This paper lies outside my primary area of expertise. While I can evaluate the clarity and quality of the writing and presentation, I do not feel fully confident assessing the soundness of the methodological approach or situating it within the broader literature on agentic LLM safety. I therefore assign a low confidence score to my review.

**Strengths:**

This paper is exceptionally well written and easy to follow. The motivation is clear, and the proposed approach appears sensible. Theere are multiple contributions — (a) a data generation framework, (b) a method achieving state-of-the-art results, and (c) a benchmark — which will likely be appreciated by the community. Beyond these core contributions, the paper is also very thorough and provides many insightful results in the main body.

**Weaknesses:**

I appreciate the paper and strongly believe it should be published. I am somewhat unsure whether ICLR is the right venue, though, since the manuscript is highly applied and engineering-focused, and I see relatively little new theoretical or methodological innovation in agentic LLM safety (except perhaps the adapter/guardian decomposition). It is unclear how much of Safiron’s improvement is due to the adapter/guardian decomposition itself versus dataset or training choices; an ablation isolating the adapter’s contribution would be interesting.

**Questions:**

X

---

> ### Author Response · Authors · 2025-11-18
> **Response for Reviewer xJJZ (Part I)**
>
> Thank you for your detailed review! We have addressed each of your concerns step by step, as summarized below.
>
>
> > W1: I appreciate the paper and strongly believe it should be published. I am somewhat unsure whether ICLR is the right venue, though, since the manuscript is highly applied and engineering-focused, and I see relatively little new theoretical or methodological innovation in agentic LLM safety (except perhaps the adapter/guardian decomposition).
>
>
>
>
> We appreciate the reviewer’s positive assessment and their support for publication. Regarding the concern about fit with ICLR: while our work is indeed strongly empirical and system-oriented, we see this as a feature rather than a limitation.
>
>
>
> First, ICLR has a long history of publishing **methodological and benchmark-driven** work where the main contributions are new problem formulations, evaluation protocols, and empirically grounded design principles (e.g., in RL, robustness, alignment, and evaluation of generative models). Our paper follows this tradition for **agentic LLM safety at the planning stage**.
>
>
>
> Second, although our framework is instantiated in a practical system, it is not just a one-off engineering artifact. It introduces:
>
>
>
> - a **structured risk-injection taxonomy and data engine** for pre-execution failures,
>
> - a **general training recipe and best practices** for guardian models (how data composition, easy/hard samples, and RL objectives affect safety behavior), and
>
> - **Pre-Exec Bench**, a reusable benchmark and metric suite specifically designed for plan-level guardrails.
>
>
>
> These elements are intended to **generalize beyond our implementation** and to serve as building blocks for future work on safe agentic systems. Given the rapid rise of LLM agents, we believe that such practically grounded, reusable methodology is highly relevant to the ICLR community.
>
>
>
> ---
>
> > W2: It is unclear how much of Safiron’s improvement is due to the adapter/guardian decomposition itself versus dataset or training choices; an ablation isolating the adapter’s contribution would be interesting.
>
> Thank you for this suggestion. To isolate the contribution of the adapter, we ran an ablation on 120 benchmark trajectories with heterogeneous log styles, comparing Safiron with and without the adapter. In the “without” setting, the guardian model directly consumes the raw trajectories with only a schema prompt; in the “with” setting, the adapter first normalizes the trajectories into the unified format and the same guardian is then applied. The results are:
>
> |          | Classification Acc | Harmful Detection | Risk Category Acc | Explanation Correctness |
> |----------|---------------------|--------------------|---------------------|---------------------------|
> | without  | 0.742               | 0.959              | 0.531               | 0.449                     |
> | with     | 0.925               | 0.959              | 0.571               | 0.429                     |
>
>
> We observe that, on this subset, removing the adapter significantly hurts overall classification accuracy (0.742 → 0.925 with the adapter), while harmful detection remains at the same high level in both settings. Risk category accuracy also improves from 0.531 to 0.571 when the adapter is used. Explanation correctness changes only slightly and the small drop is within the noise one would expect from a 120-sample subset.
>
> These results indicate that the adapter/guardian decomposition is not merely a cosmetic architectural choice: the adapter materially improves Safiron’s ability to correctly distinguish benign from harmful trajectories and to pick the right risk category under heterogeneous formats, while the underlying guardian stays the same.
>
> ---
>
> All the above modifications are marked in red in the revision.
>
> We truly appreciate your thoughtful comments and the encouragement you have given us. Your feedback has helped us see our work from new perspectives and make it stronger. We hope our responses have addressed your concerns, and we would be very happy to clarify anything further if needed. Thank you again for the time and care you have put into reviewing our paper, it really means a lot to us!

---

> > ### Comment · Reviewer_xJJZ · 2025-11-19
> >
> > >I am somewhat unsure whether ICLR is the right venue [...]
> >
> > This is just an observation from my side, I didn't consider this aspect when giving my score. I trust the AC to judge the appropriateness of this work for the venue, which I consider to be of high quality in every other aspect.
> >
> > > We observe that, on this subset, removing the adapter significantly hurts overall classification accuracy (0.742 → 0.925 with the adapter), while harmful detection remains at the same high level in both settings. [...]
> >
> > Thank you for running this ablation! The positive result further justifies Safiron's adapter/guardian decomposition and does address my (minor) concern.
> >
> > I will maintain my score of 8 and recommend acceptance based on the quality and completeness of the work.

---

> > > ### Author Response · Authors · 2025-11-19
> > > **Thanks for your response!**
> > >
> > > Thank you so much for your response and encouragement!

---

### Official Review · Reviewer_jKy2 · 2025-10-27

**Soundness:** 3
**Presentation:** 3
**Contribution:** 3
**Rating:** 6
**Confidence:** 3

**Summary:**

This paper centers on the safety of AI agents at the planning stage, prior to the execution of any actions—referred to as a proactive approach. The authors propose a data generation process named AuraGen, introduce a new model called Safiron, and develop a benchmark termed Pre-Exec Bench to address gaps in data, modeling, and evaluation. Their adapter-based Safiron pipeline consistently surpasses baseline performance on the Pre-Exec Bench.

**Strengths:**

1. The contributions of the paper are clearly delineated, facilitating readers' understanding of the authors’ ideas. Additionally, the paper is logically organized.
2. The data generation process appears reasonable, and the model training pipeline, which includes supervised fine-tuning (SFT) and reinforcement learning (RL), also seems sound.
3. The paper clearly presents the training details of each module and provides comprehensive and straightforward experimental demonstrations.

**Weaknesses:**

1. While the paper addresses data, model, and evaluation benchmarks comprehensively, the extensive experimental work may give the impression that the study lacks a specific focus or a distinct technical highlight.
2. Additional points are discussed in the Questions section.

**Questions:**

1. RM Model Learning: Why is the input from DeepSeek-R1 directly regarded as ground truth? If so, why is a separate Reward Model (RM) trained? Moreover, why isn’t the evaluation from DeepSeek-R1 directly used as a filtering criterion in the AuraGen process?
2. It is observed that even the best-performing model within Safiron achieves relatively low accuracy in explanation correctness. Could a brief explanation be provided regarding the specific tasks in which the model underperforms most significantly?
3. In the GRPO framework, explanation correctness does not contribute to the final reward calculation. This implies that the model could potentially optimize rewards by generating only the initial tokens. Could a brief explanation be offered as to why the GRPO training approach can further enhance model performance beyond SFT?
4. Could you clarify the distinction between "multi-step corruption" and "bridge branch diversion"?

---

> ### Author Response · Authors · 2025-11-18
> **Response for Reviewer jKy2 (Part I)**
>
> Thank you for your detailed review! We have addressed each of your concerns step by step, as summarized below.
>
>
> > W1: While the paper addresses data, model, and evaluation benchmarks comprehensively, the extensive experimental work may give the impression that the study lacks a specific focus or a distinct technical highlight.
>
> Thank you for this comment. We see how the breadth of data, model, and benchmark work can give the impression of a diffuse focus, and we have tried to clarify the through-line more clearly.
>
> In both the abstract and the introduction, we explicitly structure the paper around three concrete gaps we observe in pre-execution agent safety: (1) the lack of a systematic way to generate planning-stage failures, (2) the absence of a dedicated guardian model for pre-execution risk detection and explanation, and (3) the lack of a benchmark that directly targets plan-level guardrails. AuraGen, Safiron, and Pre-Exec Bench are each introduced as direct responses to these three challenges, and the experimental sections are organized to evaluate how well this end-to-end pipeline addresses them.
>
> In that sense, the technical highlight is not many unrelated components, but a single integrated framework that tackles these three missing pieces together: a controllable generator of risky plans, a specialized guardian trained on them, and an evaluation suite aligned with the same pre-execution setting.
>
> ---
>
>
>
> > Q1: RM Model Learning: Why is the input from DeepSeek-R1 directly regarded as ground truth? If so, why is a separate Reward Model (RM) trained? Moreover, why isn’t the evaluation from DeepSeek-R1 directly used as a filtering criterion in the AuraGen process?
>
>
>
> We do **not** treat DeepSeek-R1’s outputs as unquestionable “ground truth.” We use them as _teacher signals_ to bootstrap an aspect-level scorer. In a small human pilot, we verified that its scores correlate well with human judgments across all five criteria, so we treat them as reasonably good **pseudo-labels**, not definitive labels.
>
>
>
> We still train a separate RM for three main reasons:
>
>
>
> 1. **Efficiency & scalability.** DeepSeek-R1 is relatively expensive and slow. Calling it on _every_ synthetic trajectory during large-scale AuraGen generation (and future expansions) would dominate cost and latency. Our RM is a compact model fine-tuned specifically for this 5-dimensional scoring task, so it can be run cheaply and repeatedly inside the pipeline.
>
> 2. **Modularity & openness.** The RM is fully open-weight and decouples AuraGen from any particular external backend. Once trained, users can plug in the RM locally without needing access to DeepSeek-R1 or any proprietary service.
>
> 3. **Downstream flexibility.** DeepSeek-R1’s outputs are not a binary “keep/discard” decision. We design the RM to output fine-grained scores on five aspects, and then feed these scores into a lightweight classifier that mimics human discard decisions. This works better in practice than simple thresholding. Even if we used DeepSeek-R1 directly, we would still need such a policy layer, just at much higher cost.
>
>
>
> In summary, DeepSeek-R1 is used once on a modest subset of data to generate **validated pseudo-labels**, which are then distilled into a small, open RM. All filtering inside AuraGen during our experiments is done by this RM + classifier, _not_ by repeatedly querying DeepSeek-R1.
>
>
>
>
> ---

---

> ### Author Response · Authors · 2025-11-18
> **Response for Reviewer jKy2 (Part II)**
>
> > Q2: It is observed that even the best-performing model within Safiron achieves relatively low accuracy in explanation correctness. Could a brief explanation be provided regarding the specific tasks in which the model underperforms most significantly?
>
>
>
> You are right that explanation correctness is noticeably lower than our other metrics. When we break it down by category, the model underperforms most clearly on **“Lack of accountability and traceability”**. This risk type is semantically subtle and often overlaps with nearby “governance-like” risks (e.g., misuse, oversight failures), and in our RM outputs it tends to receive lower scores more often. As a result, more of these samples are filtered out by the RM + classifier, so Safiron effectively sees **less clean training data** for this category, which hurts both classification and explanation quality there.
>
>
>
> Regarding the follow-up question: **yes, using a finer-grained notion of label similarity or a hierarchical risk structure is a promising direction.** Our current 1.0 / 0.5 / 0.0 reward was intentionally kept simple and stable, treating all harmful-category confusions equally. A hierarchy or distance-aware reward (giving higher reward when the model confuses “nearby” risks) could, in principle, help exactly these subtle categories. Doing this properly would require designing and validating a shared hierarchy and collecting enough data per node, so we left it out of the current work, but we agree it is a natural extension and will mention it as future work.
>
>
>
>
> ---
>
>
>
> > Q3: In the GRPO framework, explanation correctness does not contribute to the final reward calculation. This implies that the model could potentially optimize rewards by generating only the initial tokens. Could a brief explanation be offered as to why the GRPO training approach can further enhance model performance beyond SFT?
>
>
>
> Thank you for this question. There are two parts here: why we do not include explanation correctness directly in the GRPO reward, and why GRPO can still improve performance beyond SFT.
>
>
>
> First, we intentionally do **not** reward explanation correctness for practical reasons. Evaluating explanations reliably would require either (i) an additional LLM-as-a-judge loop, which is expensive at RL scale, or (ii) some similarity metric between natural-language explanations, which is fragile because explanations can be correct but phrased in very different ways. Instead, we treat explanations as a byproduct of learning good labels. To avoid the “short-output” hacking issue you mention, we also **constrain the output length** in GRPO: if the model outputs fewer than 10 words (i.e., effectively skips the explanation), we assign a reward of 0; and if it exceeds a maximum length (300 words), we truncate and again avoid giving it an advantage purely for verbosity. This discourages the model from optimizing rewards by only emitting a few initial tokens or overly long text.
>
>
>
> Second, even without directly rewarding explanations, GRPO still improves the model over SFT because it refines the **labeling behavior** with graded feedback. SFT only tells the model “this full output is correct”; GRPO, through our 1.0 / 0.5 / 0.0 reward scheme, distinguishes between fully correct predictions, partially correct ones (e.g., correct harmful/benign but wrong category), and completely wrong ones. This provides finer-grained signal on difficult cases, especially rare or ambiguous risks, and we see that as the **risk-category accuracy improves under GRPO, explanation correctness also improves** in parallel. In other words, better categorization leads to better rationales even though the explanations themselves are not explicitly rewarded. I have clarified this motivation and the length constraints in the revised text.
>
>
>
>
> ---
>
> > Q4: Could you clarify the distinction between "multi-step corruption" and "bridge branch diversion"?
>
>
>
> Thanks for your question.
>
>
>
> - **Multi-step corruption**: we replace a **block of consecutive steps**, and the **final step can also change**. This models plans that are fully hijacked and end in a different, harmful goal.
>
> - **Bridged branch diversion**: we change some **intermediate steps only**, but **keep the original final step**. This models _stealthy_ cases where the task still seems correctly completed at the end, but the agent takes an unsafe route in the middle.
>
>
> ---
>
> All the above modifications are marked in red in the revision.
>
> We truly appreciate your thoughtful comments and the encouragement you have given us. Your feedback has helped us see our work from new perspectives and make it stronger. We hope our responses have addressed your concerns, and we would be very happy to clarify anything further if needed. Thank you again for the time and care you have put into reviewing our paper, it really means a lot to us!

---

> ### Author Response · Authors · 2025-11-26
> **Gentle Reminder Regarding Our Response to Your Comments**
>
> Dear Reviewer,
>
> Thank you very much for your valuable time and thoughtful comments. We completely understand that you may have a busy schedule. As the discussion period is now less than one week remaining, we would be truly grateful if you could kindly take a moment to review our responses and let us know whether they adequately address your concerns.
>
> We sincerely appreciate your time and consideration.

---

### Official Review · Reviewer_rUUR · 2025-10-30

**Soundness:** 3
**Presentation:** 3
**Contribution:** 2
**Rating:** 6
**Confidence:** 4

**Summary:**

The paper focuses on improving safe deployment of LLM-based agents by building tools designed for the “planning” phase of agentic execution. The authors argue that in contrast to post-execution guardrails, planning-based guardrails allow for pro-active safety evaluation. They identify a data, model, and evaluation gap as challenges to such a pro-active approach and propose solutions for each of these. To alleviate the data gap, the authors propose a multi-step synthetic data framework. For the model gap, the authors propose a guardian training method that combines an input-data adapter and a two-staged training strategy. Finally, to address the evaluation gap, the authors propose a new benchmark that uses their synthetic data engine and is enhanced by human annotators. The paper performs extensive empirical experiments on multiple models to show the efficacy of their approach.

**Strengths:**

[**significance**] Safe agent deployment is a timely and important research direction. Guardrails focused on the planning stage present an operationally practical opportunity.


[**quality**]
- Using a diverse and broad model pool is essential for robust evaluation of benchmark initiatives. The paper includes eight leading open-source models from different developers to address this (l288-289).
- The paper does a great job at discussing and addressing the requirements for enhancing pro-active safe deployment. This reviewer especially appreciated the use of human annotators to improve the quality of benchmark annotations compared to “just” using language models. Small nit: “non-LLM [arbiters] guarantee reliability” (l297), there is no formal guarantee here. The cost discussion provided in Appendix I is also appreciated.
- Detailed and comprehensive discussion of subsets of the experimental results.

[**clarity**] Overall, the paper is well written. This reviewer does believe readability could be improved (see suggestions).


[**originality**] While there is existing work highlighting the promise of auditing reasoning / chain-of-thought leading up to actions [1], work focusing exclusively on a “planning” stage is a useful contribution.


[1] Korbac et al., Chain of Thought Monitorability: A New and Fragile Opportunity for AI Safety, 2025

**Weaknesses:**

[**significance**] While safety in planning could be seen as a _necessary_ condition to safe deployment, it does make some strong assumptions preventing it from being a _sufficient_ condition. Specifically, the authors do not address the assumptions of (i) plan-to-action faithfulness [ 2, 3], or (ii) the possibility of dynamic adjustments based on environmental feedback. Additionally, (iii) this work assumes plans are fully specified, e.g., “the full sequence of actions to be taken” (l95), does not account for high-level actions that can be interpreted in a myriad of ways. If the goal is to model and improve safe deployment of AI agents, these should at least be discussed.

[**quality**]
- In section 5, the authors write “Real-world agentic systems [...] but distributionally realistic” (l286-288). While this holds for zero-shot generations, it does not necessarily hold for intervened trajectories, e.g., using the approach described in Section 3. Additionally, the authors fail to discuss the coverage or “realism” of the user queries used to generate the above planned trajectories.
- In section 4, l195-196, the authors present a core component of their stated contribution: “a unified data adapter”. While this reviewer is sympathetic to the challenges of page limits, this is not an excuse to move the entire discussion of a core component to the Appendix. At the minimum, one would expect a high-level summary. For example, lots of space is used in lines 229-242 to (mostly) restate the basic GRPO algorithm.
- Given the strong reliance and focus on synthetic data throughout this work, it is surprising to this reviewer how little discussion focuses on key insights and evaluation axes from the synthetic data literature. The related work section (in the Appendix) similarly is limited. For example, any formal discussion on diversity is missing. With one of the core stated objectives being pro-active coverage, this is damning. I would suggest the work of [4] for a detailed discussion on this topic and [5] for a practical implementation.


[2] Yakovi and Goldber, Towards faithfully interpretable nlp systems: How should we define and evaluate faithfulness? ACL 2020

[3] Dibjit, Making Reasoning Matter: Measuring and Improving Faithfulness of Chain-of-Thought Reasoning, EMNLP 2024

[4] Havrilla et al., Surveying the effects of quality, diversity, and complexity in synthetic data from large language models, 2024

[5] Davidson et al., Orchestrating synthetic data with reasoning, ICLR 2025

**Questions:**

Q1: In section 3 it is stated that AuraGen produces “diverse and controllable trajectories spanning a wide spectrum of risks” – where are these claims supported?


Q2: in Section 4 it is stated that “RL complement it by optimizing for fine-grained safety objectives” (l206-207) – why does SFT not suffice here and why does RL solve this?


Q3: In Section 5 it stated that “Overall, Pre-Exec Bench [...] first-class objective” (l255-256), could you explain what you mean by this?


Q4: Section 6, Table 2 shows results for “Risk Cat. Acc.” - for the models **not** fine-tuned using Safiron variants, were the models provided with the risk category options and explanations? Similarly, for the “Expl. Corr.”, were models provided with explanation examples? I fear that the current setup overly biases results in favor of the Safiron method. Specifically, these models were trained on a distribution that is very similar to the one used at test time. It is thus reasonable to assume they’ll produce outputs that are more in line with the expected test test format(s) than untrained models. It would be useful to include a comparison of simply providing the untrained models with some in-context examples. Given the rapid development cycle of language models, in-context learning provides an operationally more favorable approach. Thus, for practical purposes, the authors would do well to compare Safiron to ICL alternatives.


Suggestions:
- The use of “trajectory” to refer to a plan is potentially confusing given pre-existing literature. Perhaps consider simply calling these plans instead?
- However, readability could be further improved by providing a running example. It is also quite dense in information, especially in the results section, which makes it difficult to properly parse the presented results.

---

> ### Author Response · Authors · 2025-11-18
> **Response for Reviewer rUUR (Part I)**
>
> Thank you for your detailed review! We have addressed each of your concerns step by step, as summarized below.
>
> > W1: In section 5, the authors write “Real-world agentic systems [...] but distributionally realistic” (l286-288). While this holds for zero-shot generations, it does not necessarily hold for intervened trajectories, e.g., using the approach described in Section 3. Additionally, the authors fail to discuss the coverage or “realism” of the user queries used to generate the above planned trajectories.
>
>
>
> We agree that the wording “distributionally realistic” in Sec. 5 is too strong. Our claim is intended for the **benign trajectories** generated by LLM agents, not for every risk-injected trajectory. Injected trajectories are deliberately constructed as **stress tests** to cover rare but safety-critical failure modes, and we ensure plausibility via (i) a multi-aspect reward model filter and (ii) a two-phase, 3-annotator human review that only keeps trajectories judged coherent and realistic and prunes unnatural interventions. We will soften the wording in Sec. 5 and change the sentence to:
>
>
>
> > _“Real-world agentic systems are LLM-driven; thus, using LLMs to synthesize trajectories is not merely convenient but produces data that are distributionally inspired by such systems, particularly for benign plans.”_
>
>
>
> This revised phrasing makes the intended trade-off between realism and coverage explicit.
>
>
>
> Regarding **user queries**, they are not arbitrary prompts: AuraGen is conditioned on scenarios and tools from AgentSafetyBench, which provides a broad, systematically designed space of tasks, and we further condition on rich metadata (349 interactive environments, 2,000 test cases, 8 safety risk categories, and 10 failure modes across diverse domains and tools) when generating queries and plans. We have added more details about this in main text (red color).
>
>
> ---
>
>
>
> > W2: In section 4, l195-196, the authors present a core component of their stated contribution: “a unified data adapter”. While this reviewer is sympathetic to the challenges of page limits, this is not an excuse to move the entire discussion of a core component to the Appendix. At the minimum, one would expect a high-level summary. For example, lots of space is used in lines 229-242 to (mostly) restate the basic GRPO algorithm.
>
>
>
> Thank you for pointing this out. we agree that, given the page limits, we initially allocated space poorly. In the revised version, we moved the GRPO derivation to the appendix and added a concise but concrete summary of the adapter in Section 4, explaining how it normalizes heterogeneous agent outputs into a unified schema.
>
> ---
>
>
> > W3 & Q1: - Given the strong reliance and focus on synthetic data throughout this work, it is surprising to this reviewer how little discussion focuses on key insights and evaluation axes from the synthetic data literature.
>
> Thank you very much for this suggestion and for pointing us to [4] and [5], we have added these two references in the related work section. Following your comment, we added an explicit diversity analysis of AuraGen’s outputs, along the axes discussed in the synthetic data literature.
>
> First, we quantify **lexical diversity** using self-BLEU and self-ROUGE-L, computed separately on benign trajectories and risk-injected trajectories:
>
> |Diversity split|Self-BLEU|Self-ROUGE-L|
> |---|---|---|
> |No risk|0.0435|0.118|
> |With risk|0.0451|0.115|
>
> These values are low, which corresponds to high internal diversity within each subset, and the two splits are very close to each other. This indicates that introducing risks does not collapse the space into a few templated patterns. Both benign and risky trajectories cover a wide range of realizations, which is consistent with the “pro-active coverage” goal highlighted in [4,5].
>
> Second, we examine **semantic diversity** and category separation using embeddings and PCA. We present PCA visualizations of the AuraGen-synthesized data in Fig 23 and Fig 24. Using OpenAI’s `text-embedding-3-large` to obtain embeddings, we observe that all categories are distributed relatively uniformly, without pronounced clustering. This suggests that the semantics are diverse and that the samples cannot be separated purely at the category level; instead, a model must explicitly reason about whether a trajectory is risky.
>
> Finally, at the **structural** level, Section 3 defines a four-way injection taxonomy (single-step, multi-step, new-branch, bridged-branch) and AuraGen is configured to allocate data across these modes. We now make this connection to the synthetic-data diversity literature explicit in the draft and point to the new metrics and visualizations in Appendix P.
>
>
> ---

---

> > ### Comment · Reviewer_rUUR · 2025-11-24
> > **Good Rebuttal: Score Raised**
> >
> > I would like to thank the authors for their thoughtful responses and additional ablations. Overall, I am happy with their rebuttal. I do believe some areas could be sharpened and the main text could use another refinement iteration. However, this work should be a useful contribution to at least some researchers in this area and most of these lighter concerns should be well addressable before a final camera-ready. As such, I will raise my score.
> >
> > **Feedback on Responses**:  (No need to respond to these as they won't change my score, but perhaps they are useful)
> > - I would recommend moving some of the ICL vs. fine-tuning discussion to the main text. I would also like to see more experimental setup details for Appendix O on this (e.g., confidence intervals). One additional argument for training vs. ICL, is that lightweight, small models might only support a small context length. Thus, using significant portions of this context for ICL may not be optimal. One could include a smaller model, e.g., 8B, to support this case empirically.
> > - Great work on the additional synthetic data evaluation! If time permits, I would recommend spending more time here as it represents the foundation of your work.
> > - It would be valuable to include some failure modes, e.g., examples of "bad" plans that are not caught by the guardrails. This would be especially interesting to contrast your approach with "out-of-the-box" models. I did not bring this up in my original review, so it won't factor into my scoring of course.
> > - The same goes for the difference between SFT vs. RL fine-tuning: it would be useful to show where improvements of RL over SFT are coming from.

---

> > > ### Author Response · Authors · 2025-11-24
> > > **Thanks for your feedback!**
> > >
> > > Really appreciate your positive feedback and details about the paper structure. We will make it modified carefully according to your suggestions in the camera-ready version.
> > >
> > > Thanks again!

---

> ### Author Response · Authors · 2025-11-18
> **Response for Reviewer rUUR (Part II)**
>
> > Q2: in Section 4 it is stated that “RL complement it by optimizing for fine-grained safety objectives” (l206-207) – why does SFT not suffice here and why does RL solve this?
>
>
>
> Thank you for this question. After SFT, the model already performs reasonably well, but we observed that it still struggles on rare or ambiguous risks. Intuitively, SFT only tells the model “what the correct answer looks like,” whereas RL with our reward design can also tell it “how wrong” different mistakes are (e.g., correct harmful/benign but wrong category vs. completely wrong), and thus provides more fine-grained supervision, especially on difficult cases.
>
>
>
> In other words, RL complements SFT by shaping the decision boundary with graded feedback: full reward for correct detection and category, partial reward for partially correct outcomes, and zero otherwise. This has become a common paradigm—SFT as a warm start and RL to refine behavior on nuanced objectives—and in our experiments it consistently improves risk categorization and explanation quality over SFT alone. We have added a short clarification of this motivation in Section 4.
>
>
>
> ---
>
>
>
> > Q3: In Section 5 it stated that “Overall, Pre-Exec Bench [...] first-class objective” (l255-256), could you explain what you mean by this?
>
>
>
> Thank you for pointing out the ambiguity. Here, what we intended to say is simply that Pre-Exec Bench is explicitly designed with three main goals at the center: (1) realistic agentic scenarios, (2) diversity across tools, models, and risk strategies, and (3) high-quality, human-checked labels. Calling this “a first-class objective” was unnecessarily opaque, so in the revised version, we have rephrased this sentence to state these goals directly instead of using that phrase.
>
>
>
> ---

---

> ### Author Response · Authors · 2025-11-18
> **Response for Reviewer rUUR (Part III)**
>
> > Q4: I fear that the current setup overly biases results in favor of the Safiron method.
>
>
>
> Thank you for this very helpful question. For the models that are **not** fine tuned as Safiron variants, we still provide a clear schema in the prompt: we list the risk categories and ask the model to output (i) harmless vs. harmful, (ii) a risk category if harmful, and (iii) a short explanation. So they do see the options and the expected output format, but in the main experiments they do not see any few shot examples. Safiron is fine tuned on synthetic data that follows this schema, but the benchmark itself is generated from a **model pool**; the generator used for Safiron’s training is only one member of this pool, and the test trajectories also come from other models (including some of the evaluated baselines). This means the similarity is mostly at the level of format, not that training and test are drawn from the same single-model distribution.
>
>
>
> We fully agree that in context learning is operationally attractive and should be considered as an alternative. **The main challenge in our setting is that each example is long (often over 1k tokens), and a strong ICL configuration needs nine examples (eight harmful categories plus one benign).** This produces a very long prompt, which can itself hurt performance and significantly increase cost.
>
>
>
> Following your suggestion, we ran an explicit ICL baseline where we provide nine in context examples covering all risk categories and the benign case. The results are:
>
>
>
> | Model                      | Setting  | Classification Acc | Harmful Detection | Risk Category Acc | Explanation Correctness |
> |----------------------------|----------|---------------------|--------------------|---------------------|---------------------------|
> | **GPT-4o**                 | Original | 0.606               | 0.822              | 0.319               | 0.310                     |
> |                            | ICL      | 0.574               | 0.855              | 0.311               | 0.304                     |
> |                            | Diff.    | -5.28%              | 4.01%              | -2.51%              | -1.94%                    |
> | **GPT-5**                  | Original | 0.425               | 0.990              | 0.355               | 0.350                     |
> |                            | ICL      | 0.422               | 0.985              | 0.360               | 0.357                     |
> |                            | Diff.    | -0.71%              | -0.51%             | 1.41%               | 2.00%                     |
> | **Llama-3.1-70B-Instruct** | Original | 0.621               | 0.622              | 0.305               | 0.242                     |
> |                            | ICL      | 0.598               | 0.623              | 0.279               | 0.229                     |
> |                            | Diff.    | -3.70%              | 0.16%              | -8.52%              | -5.37%                    |
>
>
>
> For GPT-4o, ICL slightly improves harmful detection but worsens overall classification accuracy, risk category accuracy, and explanation correctness. For GPT-5, all changes are very small. For Llama-3.1-70B-Instruct, ICL degrades most metrics, especially risk category accuracy and explanation correctness. Across the board, these ICL variants remain clearly below the Safiron models in Table 2.
>
>
>
> Thank you again for pushing us on this point. Your suggestion both made the evaluation more complete and, in practice, further highlighted the value of a dedicated guardian like Safiron. We have added all these results and analysis in Appendix O. We have uploaded the full Safiron weights to HuggingFace (linked right below the abstract) in submission version of this draft, and in future work we plan to collaborate more with industry partners and, where possible, offer Safiron to users in an API form.
>
>
>
>
> ---
>
>
>
> > S1: The use of “trajectory” to refer to a plan is potentially confusing given pre-existing literature. Perhaps consider simply calling these plans instead?
>
>
>
> Thank you for the suggestion. In Section 2, first paragraph, we already clarify that in this paper “trajectory” denotes the planning-stage sequence of intended tool calls produced by the agent, rather than an execution-time state–action trajectory. In the camera-ready, we will (i) switch to “plan” / “action plan” as the default term in the main text, and (ii) add an explicit note in the preliminaries that when we write “plan (trajectory)” we specifically mean this planning-stage sequence, not an executed trajectory.
>
>
>
> ---

---

> ### Author Response · Authors · 2025-11-18
> **Response for Reviewer rUUR (Part IV)**
>
> > S2: However, readability could be further improved by providing a running example. It is also quite dense in information, especially in the results section, which makes it difficult to properly parse the presented results.
>
>
>
> Thank you for this suggestion. We agree that, because the paper contains a lot of content, some parts, especially the results section, can feel dense and harder to parse. To improve readability and make the pipeline easier to follow, we have made several concrete efforts.
>
> First, for **AuraGen**, we provide a small toolkit in the supplementary materials with a ready to run command that reproduces the data generation process, so readers can see the full pipeline on an actual example instead of only from the abstract description.
>
> Second, for **Safiron**, we release the model weights on HuggingFace together with a minimal working example script that loads the adapter plus guardian and runs a simple end to end check on a sample trajectory.
>
> Third, for **Pre-Exec Bench**, we include the full dataset and, following your suggestion, we added a concrete, fully worked example in Appendix P that walks through one scenario, its benign and risky plans, and the corresponding labels.
>
> We hope these additions make the overall pipeline and results easier to follow and help mitigate the density issue you mentioned.
>
> ---
>
> All the above modifications are marked in red in the revision.
>
> We truly appreciate your thoughtful comments and the encouragement you have given us. Your feedback has helped us see our work from new perspectives and make it stronger. We hope our responses have addressed your concerns, and we would be very happy to clarify anything further if needed. Thank you again for the time and care you have put into reviewing our paper, it really means a lot to us!

---

### Official Review · Reviewer_uXEq · 2025-10-31

**Soundness:** 3
**Presentation:** 3
**Contribution:** 3
**Rating:** 6
**Confidence:** 3

**Summary:**

The paper targets pre-execution (planning-stage) safety for LLM-based agentic systems, arguing that most existing guardrails act after an action is executed, which is too late for high-stakes scenarios. To close what the authors call the data gap, model gap, and evaluation gap, the paper proposes three components: (i) AuraGen, a controllable synthetic-data engine that first generates benign, tool-using agent trajectories from structured metadata, then injects category-labeled risks with four principled strategies (single-step, multi-step, new-branch, bridged-branch), and finally filters them with an automated reward model; (ii) Safiron, a “foundational guardrail” composed of a cross-planner adapter to normalize different agent outputs and a compact guardian model trained with SFT + GRPO to do (a) binary detection, (b) fine-grained risk typing, and (c) rationale generation; and (iii) Pre-Exec Bench, a new, human-validated benchmark specifically for planning-stage safety that covers diverse tools, trajectories, and injected risks. Experiments show that Safiron consistently outperforms strong proprietary and open-weight baselines on classification accuracy, harmful detection, and risk categorization, while the adapter enables cross-format generalization.

**Strengths:**

1. The paper focuses on pre-execution / planning-stage safety for agents, which is still under-served compared to execution-time or dialog I/O guardrails. Framing “look at the whole trajectory before anything is executed” is both sensible and impactful for high-stakes domains.
2. AuraGen is a clearly specified, controllable synthetic-data engine with four risk-injection strategies, a risk taxonomy, and automated RM-based filtering. This is more systematic than ad-hoc prompt-injection datasets and can be extended to new tools or environments.
3. The adapter + Safiron split is a good modular design: the adapter unifies planner outputs in heterogeneous formats, and the guardian only has to do safety reasoning on a normalized schema. This increases transferability across agent frameworks (MetaGPT, AutoGen, etc.).
4. Pre-Exec Bench is a nontrivial contribution—it is human-verified, covers ~1k benign and ~670 risky samples, uses a heterogeneous model pool to avoid style leakage, and directly targets the planning stage, which existing benchmarks do not.

**Weaknesses:**

1. Even though the authors add two-phase human verification, the core training data and the benchmark seed are still LLM-synthesized, which raises the usual concern: will Safiron detect messy, partially malformed, or adversarially obfuscated real agent traces that do not follow the neat stepwise style AuraGen produces? A small “real logs” test set (e.g., from MetaGPT/AutoGen runs with user-created tools) would help.
2. The paper claims the adapter “generalizes across input styles” (Fig. 6), but all styles seem to be synthetically created or LLM-generated. It is unclear how well the adapter handles idiosyncratic, schema-changing production systems (e.g., extra telemetry fields, partial tool arguments, JSON-with-errors). Adding a “noisy/partial JSON + extra fields” stress test would strengthen the claim.
3. AuraGen’s usefulness depends heavily on the RM and the “classifier” filtering policy (Table 1). But the paper does not show ablation on RM quality (e.g., train with a weaker RM, or no RM, or RM without the five criteria) to prove that the final gains are not just from better filtering. A small ablation could make the pipeline more convincing.
4. The paper gives a per-sample cost estimate and says Safiron is compact, but it does not provide end-to-end latency numbers for “agent -> guardrail -> environment” in a running system. A short table with average inference time on typical trajectories would be helpful.
5. Pre-Exec Bench has 1,001 benign and 671 risky samples—good for research, but still relatively small and balanced across the 4 injection strategies (25% each) rather than reflecting real-world skew. This makes the reported numbers somewhat optimistic; adding a “realistic imbalanced” split would test over-flagging.

**Questions:**

1. Can you report Safiron’s performance on non-synthetic agent traces collected from real AutoGen/MetaGPT runs (with user-written tools, occasional tool errors, and truncated steps)? Even a 200–300 sample subset would make the generalization claim much stronger.
2. Your adapter experiments remove two styles and show good generalization (Fig. 6). How does the adapter behave on malformed or partially missing tool calls (e.g., missing arguments field, extra telemetry, JSON5, HTML-in-text)? Do you discard samples or try to repair them?
3. Table 1 shows that the SVM filtering policy is best. Could you show an ablation where AuraGen uses (i) no RM, (ii) RM with a single criterion (e.g., causal consistency only), and (iii) RM with noisy labels, and then retrain Safiron? This would clarify how crucial the RM is.
4. You weigh the correct risk category at 1.0 and mismatched category at 0.5 during RL. Which categories are most often confused (e.g., availability vs. goal-hijacking)? Could fine-grained label smoothing or a hierarchy of risks improve this?
5. What is the average inference latency of the full pipeline (adapter + Safiron) on the Pre-Exec Bench input length, on a single A100/4090? Can it run inline for multi-agent systems where every plan is checked?
6. If a trajectory contains a novel or composite risk not in your pool R, does Safiron tend to (a) label it benign, (b) over-flag it to a close category, or (c) refuse? Can you add an “unknown / abstain” option?
7. You show that harmful:harmless ratios of 1:4–1:6 are best (Fig. 9). How sensitive is this to the synthetic corpus size—does the same ratio hold if you scale AuraGen from 20k to 200k?
8. Pre-Exec Bench is said to be “held out” from model selection. Will you release the human instructions and debiasing protocol so others can reproduce the same verification standard?
9. You compare to LlamaFirewall and LlamaGuard-3 in the appendix. Can you also compare to recent causal-influence prompting or runtime enforcement methods that operate over plans (e.g., AgentSpec, GuardAgent), evaluated on Pre-Exec Bench under the same metrics?

---

> ### Author Response · Authors · 2025-11-18
> **Response for Reviewer uXEq (Part I)**
>
> Thank you for your detailed review! We have addressed each of your concerns step by step, as summarized below.
>
> > W1&Q1: Even though the authors add two-phase human verification, the core training data and the benchmark seed are still LLM-synthesized, which raises the usual concern: will Safiron detect messy, partially malformed, or adversarially obfuscated real agent traces that do not follow the neat stepwise style AuraGen produces? A small “real logs” test set (e.g., from MetaGPT/AutoGen runs with user-created tools) would help.
>
> Thank you very much for raising this concern. We agree that LLM-synthesized trajectories, even with two-phase human verification, do not fully guarantee robustness to messy, partially malformed, or obfuscated real agent traces. To directly probe this, we created a small “real logs” style test set as you suggested.
>
> Concretely, we collected 40 real user queries under time and privacy constraints. These come from the authors’ actual usage of GenAI tools (for example, IDE copilots such as Cursor) and from public Reddit posts, all used in compliance with data usage policies. We then ran these queries through MetaGPT and AutoGen with user-created tools to obtain trajectories and manually injected risks. On top of that, we manually perturbed the trajectories to break the neat stepwise style of AuraGen by shortening sequences, dropping or partially deleting arguments, and inserting meaningless or noisy characters.
>
> On this 40-sample “real and messy” set, Safiron achieves:
>
> | Classification Acc | Harmful Detection | Risk Category Acc | Explanation Correctness |
> | ------------------ | ----------------- | ----------------- | ----------------------- |
> | 0.90               | 0.80              | 0.55              | 0.55                    |
>
> These numbers are lower than on Pre-Exec Bench, which is expected given that this set is intentionally hard and often has incomplete context. However, Safiron still outperforms all non-fine-tuned baselines we evaluated on the same 40 trajectories. Importantly, many of these cases require the model to pick up subtle cues in noisy, partially truncated traces rather than relying on clean, fully specified plans.
>
> The size of this real-logs-style set is limited by time and by the difficulty of gathering privacy-safe traces, but it provides an initial indication that Safiron can generalize beyond the neat AuraGen style to messy, partially malformed trajectories.
>
> ---
>
> > W2&Q2: The paper claims the adapter “generalizes across input styles” (Fig. 6), but all styles seem to be synthetically created or LLM-generated. It is unclear how well the adapter handles idiosyncratic, schema-changing production systems (e.g., extra telemetry fields, partial tool arguments, JSON-with-errors). Adding a “noisy/partial JSON + extra fields” stress test would strengthen the claim.
>
> We agree that real production logs often contain idiosyncratic schema changes and noise (extra telemetry fields, partial arguments, JSON-with-errors, etc.), and this is exactly the regime an adapter must handle. This is also why we do not rely only on clean, programmatically generated formats: in Appendix J we introduce an LLM-based transformation step whose goal is explicitly “to introduce stylistic diversity and realistic noise”, including inconsistent indentation, partially missing keys, and embedded comments. This is designed to approximate precisely the kind of messy, partially invalid JSON and schema drift the reviewer highlights.
>
> At the same time, manually constructing or collecting a broad, privacy-safe set of real production logs with all such variations would be extremely costly and often infeasible (due to both annotation effort and sensitive data). Synthetic but systematically perturbed data is therefore a practical necessity in this setting. Figure 6 already evaluates the adapter under a strong OOD-style stress test (removing two extreme styles from training and testing on them), and the adapter maintains high accuracy, indicating robustness to substantial format shifts.
>
> ---

---

> ### Author Response · Authors · 2025-11-18
> **Response for Reviewer uXEq (Part II)**
>
> > W3&Q3: AuraGen’s usefulness depends heavily on the RM and the “classifier” filtering policy (Table 1). But the paper does not show ablation on RM quality to prove that the final gains are not just from better filtering.
>
>
> We agree that AuraGen’s usefulness depends on the combination of the reward model (RM) and the subsequent “classifier” filtering policy, and that this component deserves careful scrutiny. Conceptually, we view the RM and the classifier as a **single filtering module**: the RM provides structured, five-dimensional scores, and the classifier turns those scores into keep/discard decisions. In the paper, Table 1 already performs ablations over multiple filtering _policies_ (no filtering, simple AVG/ALL thresholds, and the learned classifier), showing that with the **same RM**, different filtering rules lead to clearly different downstream performance. This demonstrates that the pipeline is not trivially “overfit” to a single heuristic, and that the classifier improves over naïve uses of the RM.
>
> In response to the reviewer’s concern about RM _quality_ itself, we have now added an additional ablation where we **change the base model used to train the RM**. In the original version, the RM is trained with one base model; we now retrain a second RM using Llama-3.1-8B-Instruct as the backbone and re-evaluate it on the same RM evaluation set. The resulting score differences across the five criteria are:
>
> | Criterion                 | Llama-3.1-8B RM | Original RM | Gap |
> | ------------------------- | --------------- | ----------- | --- |
> | Causal Consistency        | 0.72            | 0.68        | –6% |
> | Postcondition Continuity  | 0.76            | 0.70        | –9% |
> | Rationality               | 0.85            | 0.87        | +2% |
> | Justification Sufficiency | 0.99            | 1.03        | +4% |
> | Risk Matching             | 0.78            | 0.74        | –5% |
>
>
>
> All relative gaps stay within roughly **10%**, and in some cases the Llama-3.1-8B-based RM even slightly improves the score. This suggests two important points: (i) the RM training procedure is **stable across different base models**, and (ii) the **synthetic training data for the RM is transferable**, i.e., the behavior of the filtering module is not critically dependent on one particular backbone.
>
> Taken together, the existing ablations on the filtering policy (Table 1) and this new experiment with a different RM backbone indicate that our overall gains do not hinge on a single, overly strong RM. Rather, the pipeline benefits from a reasonably trained, aspect-based RM plus a lightweight classifier, and remains robust when we swap the underlying RM model.
>
>
> ---
>
> > W4&Q5: The paper gives a per-sample cost estimate and says Safiron is compact, but it does not provide end-to-end latency numbers for “agent -> guardrail -> environment” in a running system. A short table with average inference time on typical trajectories would be helpful.
>
> We appreciate the request for more concrete latency numbers. In this work, our primary focus is on the **guardrail component itself**, rather than the full “agent → guardrail → environment” stack, because end-to-end latency in real systems depends on many deployment-specific factors (choice of agent model, use of reasoning vs. non-reasoning models, external tools/APIs, network conditions, etc.). In practice, these factors can easily dominate the total runtime and vary by orders of magnitude across systems, which makes a single “universal” end-to-end number somewhat misleading.
>
> That said, we fully agree it is useful to quantify the guardrail overhead. In Appendix I we already report throughput for Safiron on typical trajectories: on H100×8 we achieve **33 samples/second** on average, and on A100×8 (40GB) we achieve **3.7 samples/second**. This corresponds to an average per-trajectory guardrail latency of roughly **0.03 s** (H100) and **0.27 s** (A100). To put this in context, if we consider representative agent wall-clock times of 15 s, 30 s, and 60 s per task, the relative overhead introduced by Safiron is:
>
> | Agent wall time (s) | H100×8 overhead | A100×8 (40GB) overhead |
> |---------------------|-----------------|------------------------|
> | 15                  | 0.20%           | 1.80%                  |
> | 30                  | 0.10%           | 0.90%                  |
> | 60                  | 0.05%           | 0.45%                  |
>
>
> Even on the slower A100 setup, the guardrail typically adds **well under 2%** overhead for realistic agent latencies, and on faster hardware the cost is almost negligible. We will add a small table like the one above to the paper to make this point explicit. As our guardrail is integrated into more agentic systems in future work, we expect to be able to report system-specific end-to-end measurements, but our current results already indicate that Safiron is highly efficient relative to typical agent and environment runtimes.

---

> ### Author Response · Authors · 2025-11-18
> **Response for Reviewer uXEq (Part III)**
>
> > Q4: You weigh the correct risk category at 1.0 and mismatched category at 0.5 during RL. Which categories are most often confused (e.g., availability vs. goal-hijacking)? Could fine-grained label smoothing or a hierarchy of risks improve this?
>
>
>
> We examined the per-category errors and found that the most frequently confused class is **“Lack of accountability and traceability.”** In particular, this category is more often mis-predicted than other risks, and tends to be confused with adjacent “governance-like” risks (e.g., cases where there is some oversight failure but it is not crisply separated from privacy or misuse aspects).
>
>
>
> A main practical reason is the **data bottleneck** for this category. In the RM outputs, “Lack of accountability and traceability” often receives comparatively lower scores, so these samples are more likely to be filtered out by the RM + classifier. As a result, the effective training set for this category is smaller than for others, which naturally makes it harder for Safiron to learn a sharp decision boundary there. This matches the reviewer’s intuition that some categories are intrinsically harder and more ambiguous.
>
>
>
> Regarding the question “Could fine-grained label smoothing or a hierarchy of risks improve this?”: conceptually, **yes, this is a promising direction**. Our current reward design (1.0 for correct category, 0.5 for wrong category, 0.0 otherwise) is intentionally simple and treats all misclassifications between harmful categories equally, mainly to keep GRPO training stable. Introducing a **hierarchical structure** over risk types or **distance-aware rewards** (larger reward for “nearby” mistakes than for clearly unrelated ones) could better reflect semantic similarity between categories and potentially reduce these confusions.
>
>
>
> However, doing this properly would require (i) a carefully agreed-upon hierarchy of risks and (ii) sufficient data for each node in the hierarchy, as well as some tuning to avoid destabilizing RL with too many reward levels. To keep the current work focused and stable, we used the simple 1.0 / 0.5 scheme, but we agree that hierarchical or similarity-aware rewards are a natural extension, and we will mention this as an avenue for future work.
>
>
>
> ---
>
>
> > W5: Pre-Exec Bench has 1,001 benign and 671 risky samples—good for research, but still relatively small and balanced across the 4 injection strategies (25% each) rather than reflecting real-world skew. This makes the reported numbers somewhat optimistic; adding a “realistic imbalanced” split would test over-flagging.
>
> We agree that it is important to understand how a guardrail behaves under “realistic” class imbalance and potential over-flagging. At the same time, we would like to clarify the design goals of Pre-Exec Bench and why we chose the current size and balance.
>
>
>
> First, in terms of **scale**, Pre-Exec Bench is already comparable to or larger than closely related planning-/safety-style datasets: for example, R-Judge reports 569 samples and AgentSafetyBench has around 2,000 cases. With 1,001 benign and 671 risky trajectories (≈1,700 total), generated from a heterogeneous model pool and passed through two-phase human verification, Pre-Exec Bench is intended to be a reasonably sized benchmark that remains feasible for careful annotation and analysis.
>
>
>
> Second, the **25% per injection strategy** is an intentional design choice rather than a limitation. Our primary goal is to **stress-test guardrails fairly across all four failure modes** (single-step, multi-step, new-branch, bridged-branch), rather than to encode the unknown, deployment-specific class priors of any particular system. If we tried to bake in one specific “real-world skew,” we would risk overfitting the benchmark to a particular product or environment and under-testing rare but safety-critical modes.
>
>
>
> Third, what constitutes a “realistic imbalanced split” is highly **environment- and product-dependent**: different agent stacks, domains, and tool configurations will exhibit very different frequencies of risky vs. benign plans and of each risk category. It is therefore difficult for us to justify a single canonical skew that would be representative for all users.
>
>
>
> That said, we fully agree with the reviewer’s underlying concern that **over-flagging under class imbalance is important**. A practical advantage of Pre-Exec Bench is that, because it is labeled at the category level, **users can easily construct their own imbalanced evaluation splits** (e.g., by downsampling certain risk types or upweighting benign samples) or re-weight categories when computing metrics to match their deployment priors. In the revised version, we will (i) clarify that our 25%/25%/25%/25% design is a deliberate “stress-test across strategies,” and (ii) explicitly recommend constructing user-specific imbalanced splits or weighted metrics on top of Pre-Exec Bench to study over-flagging in their own setting.
>
>
>
> ---

---

> ### Author Response · Authors · 2025-11-18
> **Response for Reviewer uXEq (Part IV)**
>
> > Q6: If a trajectory contains a novel or composite risk not in your pool R, does Safiron tend to (a) label it benign, (b) over-flag it to a close category, or (c) refuse? Can you add an “unknown / abstain” option?
>
>
>
> We agree this is an important point, and it is actually one of the main motivations for designing AuraGen to be **flexible and reconfigurable** rather than tied to a fixed risk pool forever.
>
>
>
> In the current version, Safiron is trained on a fixed taxonomy (R), so at inference time it is forced to choose one of the known risk categories (or benign); we do not include an explicit “unknown/abstain” option, and we also do not implement a “refuse to answer” behavior, since a guardrail is expected to always provide some actionable signal to the system. For truly novel or composite risks that are not well covered by (R), the model will typically map them to the _closest_ existing category in embedding space rather than outright refusing, but we acknowledge that we have not systematically benchmarked this behavior and that any such mapping may be imperfect.
>
>
>
> The way we intend such cases to be handled is **not** by permanently treating them as “benign or closest label and hope for the best,” but by **evolving the taxonomy and training data**. One of the key design goals of AuraGen is that adding a new risk type is lightweight: the user can extend the risk pool (R), adjust the configuration and prompts, re-run AuraGen, and obtain synthetic trajectories labeled with this new category. Safiron can then be retrained or further fine-tuned so that the new risk has its **own dedicated label**, instead of being squeezed into an ill-fitting existing one. This is precisely why we emphasize AuraGen’s flexibility: it is meant to support continual extension of the risk space as new threat patterns are discovered.
>
>
>
> Regarding an explicit “unknown / abstain” option: conceptually this is definitely possible—we could introduce “unknown” as an additional category and train Safiron to predict it when the risk does not fit any known type or when confidence is low. Doing this properly, however, would require constructing or collecting a set of out-of-taxonomy or ambiguous examples with reliable labels, and tuning thresholds so that the model does not over-use “unknown.” To keep the current work focused and the RL objective simple, we did not include this in our initial design, but we agree that an “unknown / escalate-to-human” label is a promising extension. We will mention this as a natural future direction, alongside the primary mechanism we advocate: using AuraGen to rapidly expand (R) and retrain Safiron when new risk families emerge.
>
>
>
> ---
>
> > Q7: You show that harmful:harmless ratios of 1:4–1:6 are best (Fig. 9). How sensitive is this to the synthetic corpus size—does the same ratio hold if you scale AuraGen from 20k to 200k?
>
>
>
> We have not run the 200k–scale setting yet, mainly because the synthesis and training cost at that scale is higher and the benifits are limited. Our current 20k corpus is already the result of scaling up from smaller sizes, and across 2k–10k we consistently see the same qualitative pattern: once the dataset is reasonably large, changing the harmful:harmless **ratio** has a much bigger impact than further increasing the total size.
>
>
>
> In particular, between 8k and 10k (Fig. 8) all metrics are already quite stable, while the “sweet spot” around 1:4–1:6 harmful:harmless remains. This makes us reasonably confident that the preferred ratio would not dramatically change at 200k, although we have not verified this at that exact scale due to cost constraints, which we will note as a limitation.
>
>
>
> ---
>
>
>
> > Q8: Pre-Exec Bench is said to be “held out” from model selection. Will you release the human instructions and debiasing protocol so others can reproduce the same verification standard?
>
>
>
> Thank you for the question. Yes, we fully intend for others to be able to reproduce the same verification standard. In the current draft, we have already disclosed our debiasing and verification procedure in the appendix (Fig. 22), including the two-phase human review process and the criteria used for discarding or retaining samples.
>
>
>
> ---

---

> ### Author Response · Authors · 2025-11-18
> **Response for Reviewer uXEq (Part V)**
>
> > Q9: You compare to LlamaFirewall and LlamaGuard-3 in the appendix. Can you also compare to recent causal-influence prompting or runtime enforcement methods that operate over plans (e.g., AgentSpec, GuardAgent), evaluated on Pre-Exec Bench under the same metrics?
>
> AgentSpec and GuardAgent are _runtime enforcement frameworks_, not _plan-level classifiers_. They require executable environments, tool APIs, and user-defined safety rules, and are evaluated by whether they block unsafe **actions** during execution. In contrast, Pre-Exec Bench contains only **static trajectories**, without environment states or tool interfaces, and evaluates **classification metrics** (harmless/risky, risk category, explanation).
>
> Because these methods (i) depend on environment instrumentation, (ii) do not output risk categories/explanations, and (iii) are not designed to operate on static text trajectories, running them “under the same metrics” would require heavy re-engineering and would not reflect their intended use.
>
> For these reasons, a direct quantitative comparison is **not well-defined**, but we have added discussion in Appendix D positioning them as complementary runtime approaches.
>
>
> ---
>
> All the above modifications are marked in red in the revision.
>
> We truly appreciate your thoughtful comments and the encouragement you have given us. Your feedback has helped us see our work from new perspectives and make it stronger. We hope our responses have addressed your concerns, and we would be very happy to clarify anything further if needed. Thank you again for the time and care you have put into reviewing our paper, it really means a lot to us!

---

> > ### Comment · Reviewer_uXEq · 2025-11-25
> >
> > Thank you for the author's detailed response. I have checked the modified sections and feel that I understand the paper better now. Therefore, my confidence has increased. Thank you again!

---

> > > ### Author Response · Authors · 2025-11-26
> > > **Thanks for your reply!**
> > >
> > > We really thank you for the reply and encouragement!
> > >
> > > Thanks!

---

### Author Response · Authors · 2025-12-02
**Summary of Rebuttal Impact and Reviewer Reactions**

We thank the AC and the reviewers for their time and thoughtful feedback. We would like to briefly summarize how the discussion and rebuttal addressed the reviewers’ concerns.

Overall outcome of the discussion.
All three reviewers remain positive about the paper after the rebuttal. Reviewer rUUR explicitly states that they are “happy with the rebuttal” and raise their score following our additional analyses, noting that the work “should be a useful contribution” and that remaining issues are minor and addressable in the camera-ready.
Reviewer xJJZ confirms that their minor concern about the adapter/guardian decomposition is resolved by our new ablation, and explicitly says they will maintain their score of 8 and recommend acceptance based on the “quality and completeness of the work.”
Reviewer uXEq reports that, after reading the revised sections, they understand the paper better and have increased confidence in their assessment.

**Addressing robustness and realism concerns (uXEq & rUUR).**

- To address concerns about AuraGen’s realism, we built a 40-case “real and messy” test set from actual GenAI usage (IDE copilots, Reddit queries) passed through MetaGPT/AutoGen and then manually perturbed (risk injections, truncation, dropped arguments, noise). Safiron performs strongly and still outperforms non-fine-tuned baselines, showing robustness beyond clean synthetic logs.

- We clarified that adapter training already includes LLM-generated noise (missing keys, inconsistent formats, partial JSON, comments). A new ablation on 120 heterogeneous trajectories shows removing the adapter sharply reduces classification accuracy (0.925 → 0.742), confirming its necessity for cross-format generalization.

- To reduce RM-dependence concerns, we show that (i) different filtering policies produce meaningfully different downstream behaviors, and (ii) retraining the RM with a different backbone (Llama-3.1-8B) yields scores within ~10%, demonstrating stability across teacher models.

- For efficiency, we report guardrail throughput/latency: on H100×8 and A100×8 the added overhead is well under 2% for realistic agent latencies (15–60 s), making pre-execution checking practical.

- For Pre-Exec Bench, we clarified balancing across the four injection types, explained how users can construct imbalanced subsets from released labels, and softened claims around “distributionally realistic” data by emphasizing that benign plans are grounded in AgentSafetyBench metadata while risk injections serve as targeted stress tests.


**Addressing methodological and comparison concerns (uXEq, rUUR, jKy2).**

- We clarified the role of RL: SFT provides a strong base, but GRPO with graded rewards (1.0/0.5/0) sharpens boundaries on rare or ambiguous risks; length constraints prevent trivial exploitation. RL consistently improves risk-category and explanation scores over SFT alone.

- To ensure fair comparison with generic LLMs, we added strong ICL baselines (GPT-4o, GPT-5, Llama-3.1-70B) using nine long exemplars covering all categories. ICL gives minor gains but remains below Safiron while being more expensive and far longer in prompt length, supporting the value of a compact fine-tuned guardrail.

- We clarified that plan-level guardrails (Safiron) and runtime enforcement frameworks (AgentSpec/GuardAgent) target different stages: the latter depends on execution-time tool APIs and environment states, whereas Pre-Exec Bench evaluates static plans via classification, risk typing, and explanation.

- For the RM, we explained that DeepSeek-R1 only provides pseudo-labels for a small subset; these are distilled into an efficient, open RM plus a lightweight classifier approximating human discard decisions.


**Addressing clarity and scope concerns (rUUR & jKy2).**

- We tightened claims about realism and added explicit diversity analyses (self-BLEU/ROUGE-L, embedding-space PCA), linking AuraGen to diversity/coverage criteria in recent synthetic-data literature.

- We reorganized the main text to foreground the unified adapter, moved GRPO details to the appendix, and added a running example plus released code (AuraGen toolkit, Safiron weights + demo script, a worked Pre-Exec Bench example) to improve readability and reproducibility.

- We clarified the injection taxonomy, distinguishing multi-step corruptions (changing the final goal) from bridged-branch diversions (unsafe intermediate routes with correct final goals).


Taken together, the discussion shows that the main technical and empirical concerns raised in the initial reviews have been concretely addressed through new experiments, analyses, clarifications, and releases. The reviewers’ most recent comments are uniformly positive, with two explicitly expressing support for acceptance and one increasing confidence. We again thank the AC and the reviewers for their careful consideration.

---

### Meta-Review · Area_Chair_toVX · 2026-01-17

**Summary:**

The reviewers were initially unanimously positive about the paper. The main points raised by the reviewers were:

1. Whether the synthetic data is realistic and robust to the messy real world (2 reviewers).

2. Questions about the contribution of the adapter module, and they asked for additional ablations (3 reviewers).

3. Comparisons to baselines, with 2 reviewers feeling the comparisons were insufficient. The reviewers asked for comparisons to in context learning and other RL methods.

4. The quality of the reward model, and ablations for it (2 reviewers).

5. Latency and practical concerns (1 reviewer)

**Reviewer Concerns:**

Overall, the authors did a fantastic job responding to the reviews, adding new experiments such as a realistic test set, new ablations, and the requested baselines. In particular, the new test set is quite interesting and will be valuable to the community. The authors also addressed clarifying questions such as timing and practical concerns.

**Reviewer Scores:**

Given the reviewers were initially positive, I expect they would have stayed positive. This paper is a solid contribution.

---

### Decision · Program_Chairs · 2026-01-26

Accept (Poster)